# Practical Understanding of Cancer Model Identifiability in Clinical Applications

**DOI:** 10.3390/life13020410

**Published:** 2023-02-01

**Authors:** Tin Phan, Justin Bennett, Taylor Patten

**Affiliations:** 1Theoretical Biology and Biophysics, Los Alamos National Laboratory, Los Alamos, NM 87544, USA; 2School of Mathematical and Statistical Sciences, Arizona State University, Tempe, AZ 85281, USA; 3Department of Applied Mathematics and Statistics, Johns Hopkins University, Baltimore, MD 21218, USA; 4Arizona College of Osteopathic Medicine, Midwestern University, Glendale, AZ 85308, USA

**Keywords:** observing-system simulation experiment, mathematical oncology, computational oncology, clinical application, model identifiability, prostate cancer, precision treatment, mathematical model

## Abstract

Mathematical models are a core component in the foundation of cancer theory and have been developed as clinical tools in precision medicine. Modeling studies for clinical applications often assume an individual’s characteristics can be represented as parameters in a model and are used to explain, predict, and optimize treatment outcomes. However, this approach relies on the identifiability of the underlying mathematical models. In this study, we build on the framework of an observing-system simulation experiment to study the identifiability of several models of cancer growth, focusing on the prognostic parameters of each model. Our results demonstrate that the frequency of data collection, the types of data, such as cancer proxy, and the accuracy of measurements all play crucial roles in determining the identifiability of the model. We also found that highly accurate data can allow for reasonably accurate estimates of some parameters, which may be the key to achieving model identifiability in practice. As more complex models required more data for identification, our results support the idea of using models with a clear mechanism that tracks disease progression in clinical settings. For such a model, the subset of model parameters associated with disease progression naturally minimizes the required data for model identifiability.

## 1. Introduction

Mathematical models serve an important role in the development of cancer theory and provide a framework to integrate and understand clinical data [1,2,3,4,5,6,7,8,9]. The attractiveness of mathematical models in clinical application comes from their ability to predict possible outcomes of a hypothetical treatment scenario based off of a set of mechanisms, which contrasts the black-box approaches in machine learning. Mathematical modelers assume that the relevant characteristics of an individual for treatment design can be represented by a set of parameters [10,11,12,13]. The influence of these parameters is then expressed in functional responses, which dictate the rate of each reaction within a preset model structure. The particular forms of the functional response and model structure are often borrowed from classic ecology and population studies and tested on the data of cohorts of patients, which make them a shared canvas for all patients [14]. On the other hand, the set of parameters that distinguishes the treatment outcome is unique to each individual [15,16,17]. If this unique set of parameters can be determined for a particular patient, then the model is presumed to be useful in predicting the most appropriate treatment for that patient. Thus, the concept of mathematical modeling coincides with the central idea behind precision medicine, where treatment should be formulated from the characteristics of each individual [18,19].

Mathematical models can be used to fit clinical observations and make real-time predictions of treatment outcomes. Given a sufficient amount of data for a patient and an appropriate model, one can use various statistical techniques to estimate the patient-specific set of parameters. Yet, “sufficient” is a quantity determined by model complexity. Comprehensive models, such as those found in system biology studies, are more biologically realistic, but the extra layers of complexity often hinder attempts to estimate the patient-specific set of parameters. On the flip side, simpler models with fewer parameters may not be capable of fully capturing the set of possible clinical outcomes qualitatively and quantitatively. Yet, simple models can still sometimes be too complex relative to the available data. Let us consider a thought experiment using a simple differential equation describing the birth and death processes of a population of cancer cells *x*:(1)$$x^{\prime }=\beta x-\delta x=\left(\beta -\delta \right)x=mx,$$
where β and δ are the per capita birth and death rates, respectively. We define m=β−δ. This is perhaps the simplest growth equation that can be used as a building block for a more complex model. One example of the use of this simple model is by Claret et al. [20]. In clinical settings, estimates of cancer growth are available over time, either by direct methods, such as imaging or indirect estimates from tumor proxy [21,22,23]. The clinical data would then come in pairs of $\left(t_{n},x_{n}\right)$, where $x_{n}$ is the cancer population measured at time $t_{n}$. We can assume that the model fits the data perfectly in this thought experiment. Yet, without prior knowledge of β or δ, there would be no way to determine a unique value for either parameters in this very simple model. We can only find an infinite number of pairs of β and δ that give the same value for *m*. Note that this is related, but distinct from uncertainty or sensitivity analysis, where the uncertainty in each parameter is constrained based on the error in the parameter estimation.

The above thought experiment is a toy example meant to demonstrate the concept of model unidentifiability, which heuristically implies the inability to estimate the unique patient-specific set of parameters from the data. In other words, if the model is not identifiable (relative to the available data), then it is possible to find many sets of parameters that fit the data equally well statistically. To see why model identifiability represents a great obstacle in realizing the potential of mathematical models in precision medicine, we look at a specific example by Wu et al. [24]. This example comes from a modeling study using data from a clinical trial testing intermittent androgen deprivation therapy for treatment of metastatic prostate cancer [25]. The model used in the study is a mechanistic model developed from the classic Droop cell quota model and tested against several clinical datasets [11,15,26,27]. In fact, most of the model parameters can be determined directly from information in literature [13,28], yet the model is still unidentifiable with respect to the available clinical data. Figure 1 shows that five distinct sets of parameters demonstratively capture the data well in the fitting portion, yet only one provides accurate forecast. This example illustrates why potential issues of model unidentifiability should be addressed completely prior to clinical application of mathematical models.

## 2. Materials and Methods

### 2.1. Structural and Practical Identifiability

Depend on the types of model identifiability, there are various examples and techniques to address the issues of model identifiability [29,30,31,32,33,34,35,36,37,38,39,40]. Here, we offer our perspective on this issue. Consider a dynamical system of the form: (2)$$\begin{array}{cc} x^{\prime }\left(t\right) & =f(t,x(t),u(t),\theta ), \end{array}$$(3)$$\begin{array}{cc} y(t) & =h(x(t),u(t),\theta ), \end{array}$$
where $x\left(t\right)\in R^{m}$ represents the state variables, $y\left(t\right)\in R^{d}$ represents the measurable output (e.g., the data), $u\left(t\right)\in R^{p}$ represents the (control) input vectors, for example the administered drug, and $\theta \in R^{q}$ represent the set of constant parameters. Note that while θ can contain time-varying parameters, for a biological system, this complexity can usually be avoided by using a functional representation of the time-varying parameters or explicitly modeling the underlying processes that drive the temporal changes. Thus, for simplicity, we take θ to contain only constant parameters. The general definition of model identifiability follows [29,31].

**Definition 1** (Model identifiability). *The dynamic system given by Equations (2) and (3) is identifiable if θ can be uniquely determined from the given input u(t) and measurable output y(t). If a system is not identifiable, then it is unidentifiable. Furthermore, if y(t) does not contain error, then the identifiability of the system is referred to as structural identifiability. Otherwise, it is referred to as practical identifiability.*

The first example that we gave is an instance of structural unidentifiability and the second is a case of practical unidentifiability. We expand on what it means for a dynamical model to be identifiable with respect to a measurable output. For this next example, we will look at a one dimensional logistic equation
(4)$$x^{\prime }=rx\left(1-\frac{x}{K}\right).$$

Here, *r* and *K* are the intrinsic rate of growth and carrying capacity, respectively, for the population denoted by *x*. Let $\left(r_{1},K_{1}\right)$ and $\left(r_{2},K_{2}\right)$ be two sets of parameters and assume perfect measurement y(t)=x(t). We will also assume that x(0) is known. Then, if $\left(r_{1},K_{1}\right)$ and $\left(r_{2},K_{2}\right)$ results in the same models dynamics, then
(5)$$x^{\prime }\left(t,r_{1},K_{1}\right)=x^{\prime }\left(t,r_{2},K_{2}\right)\Leftrightarrow r_{1}x\left(t\right)-\frac{r_{1}}{K_{1}}x{\left(t\right)}^{2}=r_{2}x\left(t\right)-\frac{r_{2}}{K_{2}}x{\left(t\right)}^{2}.$$

For the condition above to hold for all x(t), we must have $r_{1}=r_{2}$ and $\frac{r_{1}}{K_{1}}=\frac{r_{2}}{K_{2}}$, which implies $\left(r_{1},K_{1}\right)=\left(r_{2},K_{2}\right)$ for almost all x(0), except for possibly a set of measure zero. For instance, if x(0)=0 or *K*, then the system is at its steady state, making the aforementioned comparison obsolete. *To put it simply, if a model is identifiable, then two sets of parameters that give the same model dynamics must be identical.* This means that an identifiable model does not have potential issue in Figure 1. Instead, the uncertainty in model forecast is solely dependent on the uncertainty in the data.

We remark that the above system does not contain a control term; however, for biological systems, if the identifiability of the system without the control can be studied, then adding the control term afterward usually does not change its structural identifiability, see the example given by Eisenberg and Jain [31]. What we showed here is an example of a direct test of model identifiability, originally used by Denis-Vidal and Joly-Blanchard [41]. While the direct test method is often not used in practice, it serves as an intuitive description for model identifiability from a dynamical system perspective.

To test for model structural identifiability, software built on differential algebra theory, such as DAISY, is the gold standard [42]. However, structural identifiability does not guarantee practical identifiability, which is necessary for clinical application. The practical identifiability of a model should be studied with data, in particular using the Fisher information matrix or profile likelihood [24,31]. In these scenarios, the available data dictates the formulation of the model. However, because mathematical models are often developed independent of the data collection, modelers often must sacrifice certain realistic aspects of the model to keep it identifiable relative to the available data. Reversely, if one first builds a set of candidate models and finds out the required data for accurate model identification, then it may be possible to obtain these data during the collection process. We consider the latter case an ideal scenario, where mathematical modelers and clinicians can collaborate effectively.

### 2.2. Observing-System Simulation Experiment via Monte Carlo Method

In the ideal scenario mentioned above, Monte Carlo simulation experiment is our tool of choice to obtain the information on the data required for a model to be identifiable. First, we introduce the statistical model [43]:(6)$$y\left(t_{i}\right)=h\left(x\left(t_{i}\right),u\left(t_{i}\right),\theta \right)+\varepsilon \left(t_{i}\right).$$
where $h(x\left(t_{i}\right),u\left(t_{i}\right),\theta )$ is the measurement, θ is the vector of parameters estimated from ${\left\{y\left(t_{i}\right)\right\}}_{i=1}^{N}$ observations at time $\left\{t_{1},\ldots ,t_{N}\right\}$. Assuming no model error, then the general form of the measurement error is
(7)$$\varepsilon_{i}=h_{i}{\left(x\left(t_{i}\right),u\left(t_{i}\right),\theta \right)}^{f}\epsilon_{i},$$
with f≥0, $\epsilon_{i}$ taken to be independent and identically distributed random variables with mean 0 and variance $\sigma_{0}^{2}$. For biological application, it is reasonable to expect the measurement error to be proportional to the measurement itself, so we fix f=1, giving us a relative error. The steps of the Monte Carlo simulation method follow [29,44].

Determine the appropriate set of true parameters $\theta_{0}$ for the simulation.Numerically solve the ODE model to obtain the measurements at desired time points.Generate M sets of simulated data from the statistical model (6) and (7) with a Gaussian error structure and a chosen standard deviation $\sigma_{0}\%$ around mean 0.Fit the model to each of the M simulated data sets to obtain the parameter estimates ${\tilde{\theta }}_{i},i=1,\ldots ,M$. Here, we take M to be 200 sets.Calculate the average relative estimation error (ARE) for each element of θ as
(8)$$ARE\left(\theta_{0}^{\left(k\right)}\right)=100\%\frac{1}{M}\sum_{j=1}^{M}\frac{\left|\theta_{0}^{\left(k\right)}-{\tilde{\theta }}_{i}^{\left(k\right)}\right|}{|\theta_{0}^{\left(k\right)}|},$$
where $\theta_{0}^{\left(k\right)}$ and ${\tilde{\theta }}_{i}^{\left(k\right)}$ are the k-th element of $\theta_{0}$ and ${\tilde{\theta }}_{i}$, respectively.Repeats steps 2 through 5 with increasing $\sigma_{0}=0,5,15,25\%$.

A model is practically identifiable if the *ARE* is less than the variance (σ>0%), meaning we want the error in the parameter estimation to be less than the error in the data. When the variance σ is 0% and *ARE* is sufficiently close to 0%, then the model is considered to be structurally identifiable. We borrow the idea from the observing-system simulation experiment where we will use different hypothetical sets of data to test whether they can help us identify key parameters [45,46]. By continually restricting the amount of information we have from the data, we can approximate the threshold of information required for model identification.

The MC simulation approach is not error-free. One such limitation comes from choosing the initial guesses. For example, if we start our initial guess close enough to the true set of parameters, then the effect of the error σ may be limited. However, if we have our guesses far away from the true set of parameters, then the numerical optimization may become trapped in some local minimum away from the true estimate or the parameters may not be sensitive enough to be estimable. One can start with random samples of initial guesses to have a better chance at reaching the true estimates. However, this sampling approach does not inherently deal with the issue of the insensitive parameters. Here, we pick the initial guesses randomly within 50% of the true value. In order to rule out any parameters that are not sensitive with respect to the tolerance of the numerical optimization schemes, we carry out the MC approach for each individual parameter with error-free data ($\sigma_{0}=0\%$). Any parameters that cannot be refitted within reasonable ARE will be eliminated (or become fixed) from the pool of free parameters. The remaining parameters are deem to be sensitive enough for the numerical scheme. As we amp up the tolerance of the numerical scheme, eventually we should be able to fit all parameters when no error is present in the data.

### 2.3. Two Mathematical Models for Prostate Cancer

Many mathematical models for prostate cancer have been developed in the past two decades [1,18,47] with many recent studies focused on immuno- and chemo-treatments of prostate cancer [48,49,50,51,52,53,54,55,56]. Here, we divert from this trend and instead use two simpler mechanistic models to demonstrate the concept of model identifiability in practice. Both models contain a clear prognostic parameter that keeps track of cancer progression, which greatly simplifies their structure. Using these models, we aim to show that even if the model itself may not be identifiable, having a model-based prognostic parameter allows modelers to focus the resource to identify these key parameters. This would be helpful in practical settings due to limitation in data acquisition.

*A cancer stem cell model.* Cancer stem cells propel cancer’s therapeutic resistance and are thought to be a primary factor in the initiation and progression of prostate cancer [57,58,59,60]. Utilizing the mathematical model below, in conjunction with the stem cell hypothesis, could provide a better understanding of prostate cancer’s acquisition of castration resistant cells and their heterogeneity within a mass. Prostate cancer stem cells are thought to express little to no androgen receptors, giving them the ability to multiply their population without a hormone requirement [61]. Resistance is achieved with cancer stem cells’ ability to thrive in the absence of androgen, which provides a means for cancer to continue to evolve during and after intervention with intermittent androgen deprivation therapy [12,17,62].

Prostate cancer stem cells continue to rapidly divide after treatment, either asymmetrically to form differentiated cells or symmetrically to form additional stem cells. The production of differentiated cells results in negative feedback of the production of stem cells. However, unlike stem cells, differentiated cells are affected negatively by androgen deprivation therapy. The ability to withstand androgen deprivation is just one of the many contributing factors that give rise to the renewal of stem cells. For instance, mitochondrial fission factor expression plays a role in the evolution and multiplicity of prostate cancer stem cells [63].

Here, we use a novel model built upon this concept for prostate cancer from the studies by Brady-Nicholls et al. [12,17,62]. The model consider three compartments, the cancer stem cells (*S*), the differentiated cancer cell (*D*), and the PSA byproduct (*P*). While it is simpler in structure, the model has shown promises in its applicability.
(9)$$\begin{array}{cc} \frac{dS}{dt} & =\underset{\mathrm{growth}}{\underbrace{\frac{S}{S+D}p_{s}\lambda S}}, \end{array}$$
(10)$$\begin{array}{cc} \frac{dD}{dt} & =\underset{\mathrm{growth}}{\underbrace{\left(1-\frac{S}{S+D}p_{s}\right)\lambda S}}-\underset{\mathrm{death}}{\underbrace{\alpha T_{x}D}}, \end{array}$$
(11)$$\begin{array}{cc} \frac{dP}{dt} & =\underset{\mathrm{production}\mathrm{by}\mathrm{cancer}\mathrm{cell}}{\underbrace{\rho D}}-\underset{\mathrm{clearance}}{\underbrace{\psi P}}. \end{array}$$

The cancer stem cell population *S* is assumed to divide at a rate λ to produce either one stem cell and one cancer cell with probability $p_{s}$, or two cancer cells. This division has a negative feedback from the differentiated cancer cells, which takes the form $\frac{S}{S+D}$. The cancer cell is killed by the drug at a constant rate α, where $T_{x}$ denotes the application of the drug. PSA is produced by cancer cells at a rate ρ, which is cleared from the blood stream at a rate ψ.

Since the drug applications for these model, *u* and $T_{x}$, are known input. For simplicity, we can treat them as constant. Since they are known, their variation in time should not affect the identification of the other factors. Additionally, in practice, the drug application would be fixed for a certain period of time depending on the specific treatment. We take the following parameter values as the true values for our study: $T_{x}=0.5$ (dimensionless), $p_{s}=0.03$ (dimensionless), $\lambda =ln(2){\mathrm{day}}^{-1}$, $\alpha =0.05{\mathrm{day}}^{-1}$, $\rho =1.87\times 10^{-4}\;\mu g\;L^{-1}{\mathrm{day}}^{-1}$, and $\psi =0.085{\mathrm{day}}^{-1}$ [62].

*A cell quota cancer model.* Prostate cancer cells require androgen for growth, which is why the effect of androgen is regularly incorporated into prostate cancer model [1,64,65]. However, the quantitative connection between androgen and prostate cancer growth is not well characterized, leading to various functional forms used for this purpose.

Here, we use a cancer model that integrates the effect of androgen based on a stoichiometric modeling framework [15,66,67]. The model was developed in a series of studies that highlight the importance of androgen dynamics in prostate cancer growth [11,13,16,26,27,28,64,68,69]. In this model, cancer independence to androgen is modeled as a variable explicitly and can be used as an indicator of cancer growth. Meade et al. later expanded on this idea to build a more biologically realistic model of cancer growth for predicting treatment failure [16]. Despite its simplicity, the model is founded on established biological principle and can capture and predict the dynamics of cancer progression.
(12)$$\begin{array}{cc} \frac{dx}{dt} & =\underset{growth}{\underbrace{\mu_{m}\left(1-\frac{q}{Q}\right)x}}-\underset{death}{\underbrace{\left(\nu \left(t\right)\frac{R}{Q+R}+\delta x\right)x}}, \end{array}$$
(13)$$\begin{array}{cc} \frac{d\nu }{dt} & =-\underset{\mathrm{rate}\mathrm{of}\mathrm{gaining}\mathrm{androgen}\mathrm{independence}}{\underbrace{d\nu }}, \end{array}$$
(14)$$\begin{array}{cc} \frac{dQ}{dt} & =\underset{\mathrm{androgen}\mathrm{influx}\mathrm{to}\mathrm{cells}}{\underbrace{\left(\gamma_{1}u+\gamma_{2}\right)\left(Q_{m}-Q\right)}}-\underset{\mathrm{uptake}}{\underbrace{\mu_{m}\left(Q-q\right)}}, \end{array}$$
(15)$$\begin{array}{cc} \frac{dP}{dt} & =\underset{\mathrm{baseline}\mathrm{production}}{\underbrace{bQ}}+\underset{\mathrm{production}\mathrm{by}\mathrm{cancer}\mathrm{cells}}{\underbrace{\sigma xQ}}-\underset{\mathrm{PSA}\mathrm{clearance}}{\underbrace{\epsilon P}}. \end{array}$$

The cancer population, denoted by *x*, grows based on the Droop cell-quota model. The death rates are contributed by an androgen dependent term, $\nu \left(t\right)\frac{R}{Q+R}x$, and a density dependent term, $\delta x^{2}$. Here, ν(t) is the maximal androgen dependent death rate for the cancer. The authors assume that the cancer cells lose their dependence on androgen at a rate −dν, which can be interpreted as the “rate of gaining androgen independence”. With this interpretation, under androgen deprivation therapy, the treatment would gradually become ineffective. *Q* and *P* are the intracellular androgen level and serum PSA, respectively. The dynamics of *Q* is governed by an influx of serum androgen and the uptake of cancer cells. $\gamma_{1}$ and $\gamma_{2}$ represent the rates at which androgen is being produced by the testes and the adrenal gland, respectively, with the drug application denoted by *u*. *P* is assumed to be produced as a baseline by normal cells, but mainly by cancer cells, and is cleared from the blood stream at a constant rate. We take the following parameter values as the true values for our study: u=0.5 (dimensionless), $\mu_{m}=0.009\;{\mathrm{day}}^{-1}$, $q=0.4\;\mathrm{nmol}\;{\mathrm{day}}^{-1}$, $R=3\;\mathrm{nmol}\;L^{-1}$, $\delta =45\;L^{-1}{\mathrm{day}}^{-1}$, $d=0.0001\;{\mathrm{day}}^{-1}$, $\gamma_{1}=0.08\;{\mathrm{day}}^{-1}$, $\gamma_{2}=0.004\;{\mathrm{day}}^{-1}$, $Q_{m}=30\;\mathrm{nmol}\;L^{-1}$, b=0.0001$\mu g\;{\mathrm{nmol}}^{-1}{\mathrm{day}}^{-1}$, σ=0.001$\mu g\;{\mathrm{nmol}}^{-1}L^{-1}{\mathrm{day}}^{-1}$, and $\epsilon =0.1\;{\mathrm{day}}^{-1}$ [13,15].

### 2.4. Parameter Optimization

When the data are of a single type, we use the standard root mean squared error (RMSE), for example, the cancer stem cell model with PSA data. When the data composed of multiple types of data, for example with PSA and androgen, we weigh the error contribution from each source equally. Any variation from this fitting procedure will be mentioned on a case-by-case basis. Finally, we use the built-in function *lsqnonlin* (MATLAB) for our optimization.

## 3. Results

### 3.1. The Identifiability of Two Prostate Cancer Models

First, we study the identifiability of the model given the measurements that are usually available directly for parameter estimation. In the case of the cancer stem cell model, this measurement is taken to be PSA. In the case of the cell quota model, the measurements are PSA and androgen. We also note that the spacing between the synthetic data points is kept constant in this section.

*The cancer stem cell model.* An example of the synthetic data and fitting for the cancer stem cell model is presented in Figure 2. Table 1 shows the results from the sensitivity test for each individual parameter for the cancer stem cell model. Out of the seven parameters and initials, $p_{s}$ and D(0) are not sensitive enough to be identifiable for larger measurement error. On the other hand, λ,α,ρ,ψ, and S(0) appear to be sufficiently sensitive. Thus, we fix $p_{s}$ and D(0).

Next, we carry out the MC scheme for all of the remaining parameters at once, namely λ,α,ρ,ψ,S(0). The results in Table 2 show that only ψ is practically identifiable. To see why the other parameters are not identifiable with only PSA data alone, we fix ψ and test the identifiability of all 2-combinations of the remaining parameters (e.g., fit two parameters at a time while fixing the rest). We find that none of the 2-combinations are practically identifiable. Since each parameter being tested is sensitive enough to be identifiable by themselves, this indicates the existence of an unknown relationship among the remaining four parameters (e.g., λ,α,ρ,S(0)). To demonstrate this point, we plot the estimated values of these parameters in 2-combinations and show that an approximate relationship between these parameters can be obtained by a simple regression Figure 3. Without error ($\sigma_{0}=0\%$), the relationships between λ,α, and ρ are evident by performing the 2-combination test.

In Brady et al. [62], the authors find that the prostate cancer stem cell renewal rate $p_{s}$ is a good indicator of resistance timing. However, our analysis shows that in order to utilize $p_{s}$ to make clinical predictions, modelers must have a solid grasp on the values of all other parameters and a good understanding of the appropriate value for $p_{s}$ in the cancer stem cell model. This agrees with the approach taken in Brady et al. to obtain model identifiability for $p_{s}$ [17,62].

*The cell quota cancer model.* An example of the synthetic data and fitting for the cancer stem cell model is presented in Figure 4 Similarly, we carry out these tests with respect to the 13 parameters and initials for the cell quota cancer model. Table 3 shows that out of these, only $\mu ,Q_{m}$, and ϵ are sufficiently sensitive when each parameter is fitted individually with PSA and androgen data.

When fitting the three sensitive parameters together, we find that only ϵ is practically identifiable, see Table 4. To see why the remaining two parameters (μ and $Q_{m}$) are not identifiable, we fix ϵ and study the correlation between these two parameters. Here, we find a similar linear relationship between estimates for μ and $Q_{m}$ with or without error in the data, see Figure 5. This hidden correlation interferes with the estimability of these two parameters.

In Baez and Kuang, the parameter *d* (or variable ν(t)) is created to keep track of the development of cancer resistance. However, our analysis indicates a similar issue to the cancer stem cell model where the relevant parameter is not identifiable with the available data. If we want to have an accurate estimate of *d*, we must have a strong grasp on the values of all other variables in the model and a good guess for an appropriate value for *d*.

### 3.2. Observing-System Simulation Experiment—Identifiability of Treatment Resistance Parameter

Now, we turn our focus to answering the question: what amount of data are necessary to determine the key model-based prognostic parameter? To address this question, we synthesize candidate sets of data that vary in the type and frequency of data collected. Then, we attempt to study the identifiability of these key parameters using each synthetic dataset.

*The identifiability of $p_{s}$ in the cancer stem cell model.* Recall that $p_{s}$ is not identifiable even when being estimated by itself with only PSA data (Table 1). Thus, we will carry out simulation to determine the amount of data required to sufficiently characterize the $p_{s}$, which is used to predict treatment success and failure in Brady et al. [17,62]. For the experiment, we assume all parameters (except for $p_{s}$) can be obtained from other means, which means they are fixed to the values used to make the synthetic dataset for these simulation experiments.

Table 5 summarizes the main results of the experiments to determine the identifiability of $p_{s}$. The frequency of data points appears to be the most influential factor to identify $p_{s}$, which is followed by the inclusion of the measurement of cancer stem cells. On the other hand, measurement of the cancer population, optimization tolerance, and (linear) weight contribution from different sources of error have less of an impact on the identifiability of $p_{s}$. Interestingly, if a measurement of PSA can be taken roughly every 5 h, then we could accurately determine $p_{s}$ as well (given that it is the only parameter we need to estimate).

While the results in Table 5 suggest the frequency of measurements plays a key role, when fitting all parameters together with pseudo-continuous measurement of PSA and cancer stem cells, the model remains to be unidentifiable (see Table 6). However, if the measurements are very precise ($\sigma_{0}\approx 0\%$), estimated values of each parameter are within acceptable ranges that may still be useful in making predictions (less than 10% difference from the true value).

*The identifiability of d in the cell quota cancer model.* Similarly, recall that *d* is not identifiable in the cell quota cancer model even when being estimated by itself with both androgen and PSA data (Table 3). Hence, we carry out simulation to determine the amount of data required to identify $p_{s}$. As before, we fix all other parameters for these simulation experiments.

Table 7 summarizes the main results of the experiments. We reach a similar conclusion that the frequency of data points appears to be the most influential factor in identifying *d*, which is followed by the inclusion of the measurement of cancer cells. However, with larger error margins, the measurement of cancer cells loses its effectiveness, which is problematic due to the fact that accurate estimations of cancer populations are difficult in practice. Meanwhile, all other factors have a negligible effect on the estimation of *d*. Finally, if a measurement of cancer cells, androgen, and PSA can be taken every 24 h, then we could accurately determine *d*. Interestingly, if a measurement of the cancer population is not available, but we can obtain pseudo-continuous data for *Q* and *P*, then it is possible to determine the value of *d* within a reasonable range.

As before, if none of the other parameters are known, then the identifiability of the model is not possible even with pseudo-continuous data of the cancer population, androgen, and PSA (see Table 8). Yet, if those measurements can be taken very precisely ($\sigma_{0}\approx 0\%$), then the parameters can still be estimated within reasonable accuracy for application.

## 4. Discussion

Mathematical models not only contribute to the foundation of cancer theory but can also be integrated to provide a better prognostic tool for clinicians in clinical settings. For example, one may apply mathematical models to better understand cancer progression dynamics and make predictions of treatment outcomes based on a patient’s characteristics [18]. Yet, the issue of practical identifiability remains a major obstacle to realizing the clinical potential of mathematical models. In this study, we explore the issue of model identifiability from a clinical perspective. First, we study the general identifiability property of the model. Then, we narrow down the parameter that can be used to predict treatment outcomes and look for the appropriate set of data for its identification using Monte Carlo simulation. Our results provide insights into the type of data acquisition that can enable future incorporation of mathematical models into clinical applications.

*The frequency of data collection plays a major role in model identifiability.* It is well known that increasing the number of measurements increases the chances of obtaining true estimates of model parameters assuming that the measurements are perfect and the model is structurally identifiable [70]. Here, we demonstrate in both examples that increasing the frequency of measurements, even in the presence of Gaussian noise, can increase model identifiability. However, simply increasing the number of data points will not overcome the issue of structural identification. The dataset should cover multiple temporal regions of cancer growth, so that the model can be tested more comprehensively to prove its usefulness. This finding suggests the development of devices or procedures to obtain measurements, such as PSA and androgen, on a regular basis can help to accurately identify the values of prognostic parameters.

*Cancer population data can help reduce the uncertainty in model identifiability.* We demonstrate that the inclusion of cancer population measurements (or stem cells) can increase the model identifiability. In practice, this can be completed by using imaging data or indirectly measuring circulating cancer cells [71,72,73,74,75]. On the other hand, for models that incorporate mechanisms for cancer growth using androgen, androgen data seem to be necessary for model identifiability. Unfortunately, these measurements are not widely adopted, making it difficult to integrate these ideas effectively. Furthermore, we carried out the same computational experiments on several cancer models with multiple subpopulations (not shown). The results suggest a measurement that helps to distinguish different cancer subpopulations (e.g., the frequency of each cancer subpopulation) may be necessary to obtain model identification.

*Highly accurate data may be the key to addressing model identifiability in practice.* Perhaps the most intriguing finding is that with very accurate data, the prognostic parameters and some other model parameters are reasonably identifiable. Unlike the frequency of measurements, the accuracy of measurements does not require additional compliance from the patients. With continual advances in the techniques and equipment to measure the relevant biomarkers, high-accuracy data may be the key to obtaining model identifiability in practice. We also note that certain biological markers, such as androgen, vary significantly throughout the day and with diets [64], so better clinical protocols may need to be implemented to obtain more accurate measurements.

So far, we have only discussed the applicability of model identifiability in terms of key prognostic parameters. However, one can analytically derive treatment outcomes based on a combination of a set of model parameters with mathematical analysis. This can provide a deeper understanding of key factors that drive the progression of cancer and may even shed light on novel treatments. However, to apply analytical results in practice, one needs to assess the interconnection between the parameters and how they may change based on external factors. If how these parameters change over time during treatment can be assessed, one can then use the analytical condition to determine the treatment outcome directly. Nevertheless, the issue of model identifiability remains a crucial component for this approach to work.

Another aspect of model identifiability is the statistical method used for parameter estimation. Most approaches in literature use an individual fitting, which limits the amount of data used for parameter estimation for each individual data. An alternative approach is to use population fitting with mixed effects. This should not be confused with pulling individual data and fit to the average. Instead, this approach assumes that for each parameter, its value varies per individual, but follows some distributions for the whole population. Thus, we can utilize the data of all patients simultaneously to fit the model. A software often used to implement this approach is Monolix [76]. Examples of this approach can be found in within-host viral dynamics literature [77,78]; however, it has yet to gain traction in mathematical and computational oncology literature.

In summary, we find that incorporating frequent data measurements, different types of data (especially those related to the cancer population), and high-accuracy measurements will increase the likelihood of practical identification of prognostic parameters. As more complex models contain more parameters, making it a more difficult task to obtain complete model identification, our results advocate for the use and development of models with a mechanism that tracks disease progression. By incorporating such a mechanism, a subset of model parameters (associated with the mechanism) naturally becomes the focus of model identifiability. This reduces the issue of model unidentifiability and provides a means for making predictions regarding the outcome of treatment. There are several major limitations to our studies, such as the assumption of a perfect model (no model error). These issues can perhaps be accounted for by continually improving the model development or by using a data assimilation approach, such as the Kalman filter [79,80]. We also do not employ patient-specific data for our simulation study. These can be explored in future studies.

## Figures and Tables

**Figure 1 life-13-00410-f001:**
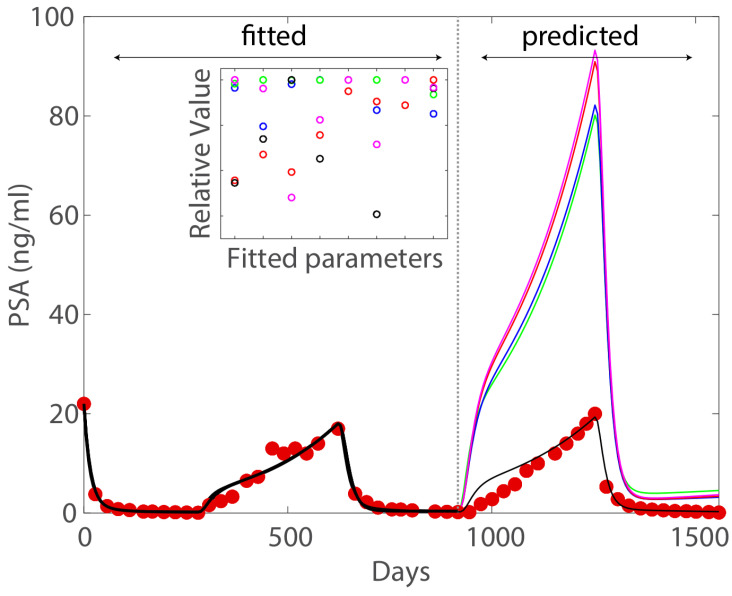
Figure adapted from Wu et al. [24] with permission distributed under a Creative Commons Attribution (CC BY) license. The color of the fitted parameters corresponds to the forecast trajectory of the same color. In the fitting portion, five different sets of parameters produce nearly indistinguishable good fits to the data. However, in the forecasting portion, only one set provides accurate forecasting.

**Figure 2 life-13-00410-f002:**
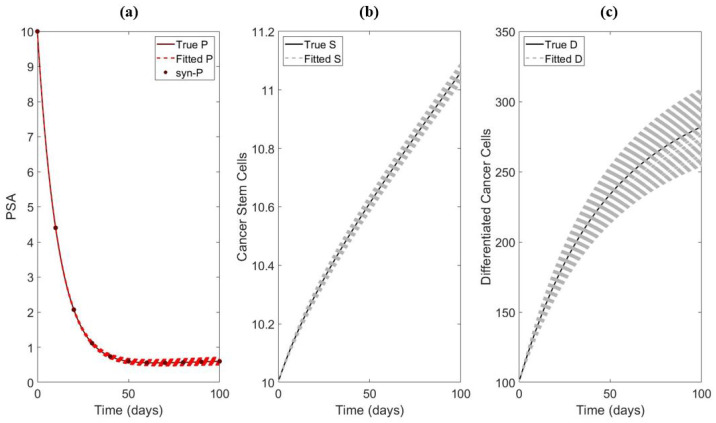
An example of data fitting with only PSA synthetic data. In this example, the parameters being fitted are λ and ρ with $\sigma_{0}=0$. (**a**) Fitting of PSA. (**b**) Simulation of the S using the best fitted parameters. (**c**) Simulation of D using the best fitted parameters.

**Figure 3 life-13-00410-f003:**
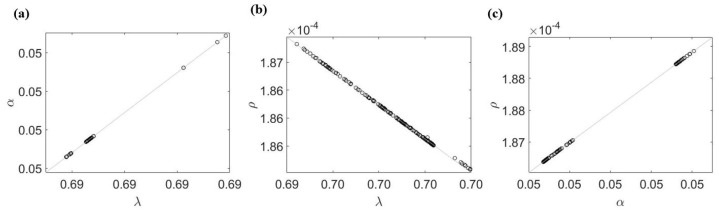
Parameter relation obtained from the 2-combination parameter test for the case of $\sigma_{0}=0$. (**a**) α and λ are positively correlated. (**b**) ρ and λ are negatively correlated. (**c**) α and ρ are positively correlated. If the parameters are linearly correlated, (**a**,**b**) would imply that α and ρ are negatively correlated; however, this is not the case in (**c**). This suggests all three parameters are involved in a non-linear relationship.

**Figure 4 life-13-00410-f004:**
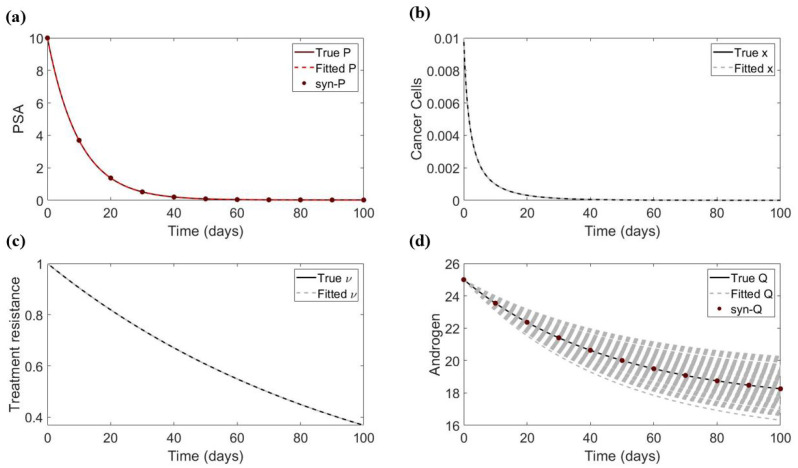
An example of data fitting with PSA and androgen synthetic data. In this example, the parameters being fitted are μ and $Q_{m}$ with $\sigma_{0}=0$. (**a**) Fitting of PSA. (**b**) Simulation of the cancer population using the best fitted parameters. (**c**) Simulation of the parameter ν (associated with treatment resistance) using the best fitted parameters. (**d**) Fitting of androgen.

**Figure 5 life-13-00410-f005:**
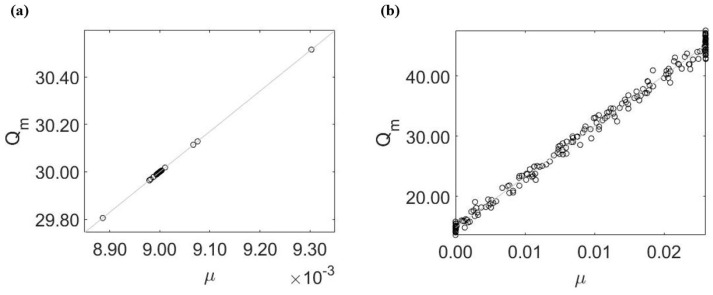
Parameter relation obtained from the 2-combination parameter test. (**a**) When $\sigma_{0}=0$, μ and $Q_{m}$ are slightly correlated with the best fit values concentrated around $\mu =9.00\times 10^{-3}$ and $Q_{m}=30$. (**b**) When $\sigma_{0}=5\%$, the relationship between μ and $Q_{m}$ is much clearer, leading to a larger ARE%.

**Table 1 life-13-00410-t001:** ARE% calculated for Brady et al. model with respect to σ. The test is carried out individually for each parameter and initial. 200 samples are used. The results indicate that five (λ,α,ρ,ψ,S(0)) out of seven components are sufficiently sensitive to to the numerical optimization.

$\sigma_{0}$	0%	5%	15%	25%
$p_{s}$	0.028	37.72	92.89	130.77
λ	0.01	2.17	6.51	10.85
α	0.01	2.94	8.89	15.20
ρ	0.01	2.50	7.49	12.48
ψ	0.01	2.12	6.61	11.89
S(0)	0.01	2.18	6.54	10.90
D(0)	0.20	35.95	87.01	121.86

**Table 2 life-13-00410-t002:** ARE% calculated for Brady et al. model with respect to σ using the MCMC method. The test is carried out for all five parameters and initial together. In total, 200 samples are used. The results indicate that when fitting together, only ψ is practically identifiable.

$\sigma_{0}$	0%	5%	15%	25%
λ	54.79	246.64	243.88	220.03
α	52.51	131.37	123.11	131.75
ρ	50.59	130.77	200.50	206.93
ψ	1.12	6.36	13.98	21.11
S(0)	13.66	37.48	70.88	101.84

**Table 3 life-13-00410-t003:** ARE% calculated for BK 1 model with respect to σ using the MCMC method. The test is carried out individually for each parameter and initials. In total, 200 samples are used. The results indicate that 3 ($\mu ,Q_{m},\epsilon$) out of 13 components are sufficiently sensitive to continue with the procedure. Note that δ and σ appear to be insensitive, so we exclude them.

$\sigma_{0}$	0%	5%	15%	25%
μ	0.00	4.48	13.49	22.68
*q*	5.78 $\times \;10^{-5}$	86.18	94.87	97.21
*R*	0.51	68.42	68.05	67.87
δ	5.02	5.02	5.02	5.02
*d*	11.70	66.07	65.74	65.91
$\gamma_{1}$	2.39 $\times \;10^{-4}$	10.89	32.76	51.04
$\gamma_{2}$	0.00	21.73	57.28	73.29
$Q_{m}$	2.28 $\times \;10^{-5}$	2.27	6.81	11.35
*b*	5.51	32.40	46.65	50.32
σ	1.05	14.94	14.96	15.97
ϵ	1.82 $\times \;10^{-4}$	2.34	7.33	13.24
x(0)	0.57	59.84	60.14	59.75
v(0)	0.46	78.75	78.57	78.29

**Table 4 life-13-00410-t004:** ARE% calculated for BK 1 pop model with respect to σ using the MCMC method. The test is carried out for all five parameters and initials together. In total, 200 samples are used. The results indicate that when fitting all four parameters together, only ϵ is practically identifiable. It is worth pointing out that when the measurement error is 0, the ARE% of all four parameters are very close to 0, indicating structural identifiability.

$\sigma_{0}$	0%	5%	15%	25%
μ	0.02	64.85	87.60	92.13
$Q_{m}$	0.01	32.77	43.91	45.80
ϵ	0.00	2.34	7.32	13.23

**Table 5 life-13-00410-t005:** The identification of $p_{s}$ in the cancer stem cell model. The test is carried out for $p_{s}$ (all other parameters and initials are fixed to their true values). Baseline frequency (data) indicates a measurement is taken every 10 days. Pseudo-continuous data indicates a measurement is taken roughly every 2.4 h. Increased optimization tolerance refers to one fold increase in the function tolerance and optimality tolerance of the optimization function. Weight (ω) comes from the minimization objective. ω>0.5 means higher weight is given to the error in *P* and ω<0.5 means higher weight is given to the error in *S*, which is given by $\omega \times RMSE^{P}+\left(1-\omega \right)\times RMSE^{S}$. Asterisk (∗) indicates practical identifiability.

$\sigma_{0}$	0%	5%	15%	25%
Fitting using P (baseline frequency)	0.03	37.72	92.89	130.77
Fitting using S and P (baseline frequency)	0.00	19.45	55.43	81.93
Fitting using D and P (baseline frequency)	0.00	30.33	79.48	113.78
Fitting using S, D, and P (baseline frequency)	0.00	29.47	77.71	111.34
Fitting using S and P (baseline frequency × 2)	0.00	13.38	39.08	60.22
Fitting using S and P (baseline frequency × 10)	0.00	6.15	18.45	30.60
Fitting using S and P (pseudo-continuous data) (∗)	0.00	1.86	5.57	9.29
Fitting using P (pseudo-continuous data) (∗)	0.00	3.69	11.05	18.42
Fitting using P (baseline frequency × 50)	0.00	5.53	16.58	27.62
Fitting using S and P (increased optimization tolerance)	0.00	19.45	55.43	81.93
Fitting using S and P (weight = 0.4)	0.00	19.43	55.38	81.87
Fitting using S and P (weight = 0.25)	0.00	19.41	55.34	81.82

**Table 6 life-13-00410-t006:** ARE% calculated for the cancer stem cell model with respect to σ. The experiment is carried out using pseudo-continuous data of *S* and *P*.

$\sigma_{0}$	0%	5%	15%	25%
$p_{s}$	1.97	34.68	97.35	107.28
λ	6.46	90.26	154.45	167.19
α	2.20	30.27	53.20	72.75
ρ	5.07	37.08	60.01	70.64
ψ	0.06	0.86	1.95	3.77
S(0)	0.02	0.43	0.93	1.44
D(0)	4.83	11.26	17.68	34.10

**Table 7 life-13-00410-t007:** The identification of *d* in cancer cell quota model. The test is carried out for $p_{s}$ (all other parameters and initials are fixed to their true values). Baseline frequency (data) indicates a measurement is taken every 10 days. Pseudo-continuous data indicates a measurement is taken roughly every 2.4 h. Increased optimization tolerance refers to a one-fold increase in the function tolerance and optimality tolerance of the optimization function fmincon (MATLAB). ${\mathrm{Weight}}_{1}=\omega_{1}$ and ${\mathrm{weight}}_{2}=\omega_{2}$ come from the minimization objective, which is $\omega_{1}\times RMSE^{P}+\omega_{2}\times RMSE^{Q}+\left(1-\omega_{1}-\omega_{2}\right)\times RMSE^{X}$. Asterisk (∗) indicates practical identifiability. (∗∗) indicates *d* is identifiable at very high error.

$\sigma_{0}$	0%	5%	15%	25%
Fitting using Q and P (baseline frequency)	11.70	66.07	65.74	65.91
Fitting using X, Q, and P (baseline frequency)	0.09	24.12	50.96	60.37
Fitting using X, Q, and P (baseline frequency × 10)	0.10	6.34	19.73	27.75
Fitting using X, Q, and P (baseline frequency × 50) (∗)	0.10	3.90	8.23	12.29
Fitting using X, Q, and P (pseudo continuous data) (∗)	0.08	2.78	6.78	10.79
Fitting using Q and P (pseudo continuous data) (∗∗)	3.67	10.44	10.69	10.75
Fitting using X, Q, and P (increase optimization tolerance)	0.09	29.04	62.88	74.00
Fitting using X, Q, and P (${\mathrm{weight}}_{1}=0.4$, ${\mathrm{weight}}_{2}=0.33$)	0.11	25.33	46.66	52.43
Fitting using X, Q, and P (${\mathrm{weight}}_{1}=0.25$, ${\mathrm{weight}}_{2}=0.25$)	0.09	19.90	43.54	52.83
Fitting using X, Q, and P (${\mathrm{weight}}_{1}=0.4$, ${\mathrm{weight}}_{2}=0.2$)	0.10	23.33	46.81	54.39
Fitting using X, Q, and P (${\mathrm{weight}}_{1}=0.25$, ${\mathrm{weight}}_{2}=0.5$)	0.10	21.26	41.25	45.78

**Table 8 life-13-00410-t008:** ARE% calculated for the cell quota cancer model with respect to σ. The experiment is carried out using pseudo-continuous data of x,Q, and *P*.

$\sigma_{0}$	0%	5%	15%	25%
μ	6.55	5.41	11.41	19.38
*q*	0.75	2.53	1.53	1.69
*R*	4.18	3.90	4.53	5.46
δ	5.17	4.48	4.97	4.90
*d*	0.04	12.78	14.30	18.69
$\gamma_{1}$	5.83	11.79	14.98	23.74
$\gamma_{2}$	3.36	15.88	11.63	14.00
$Q_{m}$	4.81	2.66	3.71	5.56
*b*	0.44	9.57	27.39	42.59
σ	0.03	26.09	34.05	39.14
ϵ	0.01	0.34	1.02	1.68
x(0)	0.37	1.65	4.30	6.84
v(0)	1.71	3.97	4.37	5.25

## Data Availability

All codes will be made available upon request.

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
