# Peer review of "Practical Understanding of Cancer Model Identifiability in Clinical Applications"

_life, 2023, doi:10.3390/life13020410_

Round 1

Reviewer 1 Report

The authors present a work that approaches mathematically the issue of model identifiability, how it can be applied in cancer modeling and how a tradeoff between structural identifiability and practical identifiability is realistic towards clinical applications. The work is interesting posing a question regarding a developed model for a disease such cancer can provide accurate predictions etc. Although the mathematical part seems legit and interesting the biological part is lacking in the ways it is presented but probably this may arise due to the small (?) experience of the authors with clinical applications. (My apologies if I am wrong). My biggest concern is that the authors choose to use a writing approach that is more often observed in computer science than in medical science community. They often mix theory with methods or results and instead of a background introduction that states the problem they provide a thought experiment. That makes the work very general and creates another problem. There are not general models in cancer. There are several cancer models in form of deterministic or stochastic with several components to describe the behavior of cancer cell growth, proliferation and response to treatment. For example the work of Yin A. et al., Pharmacometrics Syst Pharmacol. 2019 Oct;8(10):720-737. (Not cited) reviews cancer models. We do not see any introductory discussion regarding the "problem" behind cancer models and how identifiability can solve it. Moreover the authors present their work based on 2(!) models of prostate cancer which have specific characteristics. What about models for other solid tumors or leukemia etc. Finally, for a research in clinical environment that works with a treatment response tumor growth model what he/she would need? A practical identifiable model or a structural identifiable one? What about researcher in industry? What is required in terms of identifiability? Maybe if the authors had approached the problem like that would be better presented. Finally by incorporating already available models but with no means to improve them it subtracts some of the novelty.

Some comments follow:

1)      Please update title stating that you refer in cancer models (and probably models that are based on ODEs)

2)      Abstract and text in general. The authors discuss generally cancer models. I believe they mean tumor growth models that needs to be clarified and also bear in mind that this has nothing to do with disease progression in clinical level.

3)      Line 3 "In modeling studies, an individual’s characteristics are represented as parameters in a model and used to explain, predict, and optimize treatment outcome". Is it true? What about in vitro/in situ or xenograph based models (the majority of cancer models).  

4)      Lines 25-36. Too many statements with lack of supportive references

5)      Reference 5. Nom, please use published works and not works that are in pre-prints. This is fundamental. Also (from Nature portofolio) personalized medicine is a therapeutic approach involving the use of an individual’s genetic and epigenetic information to tailor drug therapy or preventive care. The term nowadays that tend to be catholic is precision medicine. According to NCI Dictionary of Cancer Terms: Precision Medicine is a form of medicine that uses information about a person’s own genes or proteins to prevent, diagnose, or treat disease. In cancer, precision medicine uses specific information about a person’s tumor to help make a diagnosis, plan treatment, find out how well treatment is working, or make a prognosis. Examples of precision medicine include using targeted therapies to treat specific types of cancer cells, such as HER2-positive breast cancer cells, or using tumor marker testing to help diagnose cancer. Also called personalized medicine.

6)      Lines 36-37 "Clinical observations form the bridge between the mathematical models and application" No they do not. The authors would be more familiar than me with Dr Box "all models are wrong but some are useful". MODELs are developed to allow us to describe, predict, study and explain how the applied medical science leads to specific observations or specific observations will lead to specific application.

1.       Clinical observations are the basic part of evidence that is needed in medicine, followed by RCTs etc. etc. If models were always needed how the anticancer medicines were found prior to digital revolution and model development? This statement contributed significantly for major revisions. If you want to sustain it provide references to support it but it is totally wrong phrase. After all as the work of Altrock, P [Nat Rev Cancer 15, 730–745 (2015)] characteristically states "Mathematical models have become an integral part of cancer biology. They are useful tools for deriving a mechanistic understanding of dynamic processes in cancer." Interestingly this work is not cited.

7)      Line 41. References! What image analysis can feed the eq 1? Thought experiment or not approach, it should contain a priori information. Any image analysis that counts (?) tumor volume or cancer cells or cell growth? Please provide the relative references.

8)      Please provide a relative graphical abstract of your work.

9)      Figure 1. 3 sets of data, where are the other 2?

10)   Lines 166-172 is discussion not methods

11)   Similarly lines 245-250 is discussion. Also the phrase "This makes sense biologically" is not proper. If it does not then the model can be mathematically right but not applicable in clinical setting. After all for what else reason is it developed?

12)   Line 313. How? What is the goal for clinical application of a model? Refs?

13)   Discussion has too many references. The issue that this works looks more like a computer science article over medical one. But this issue can be overlooked.

14)   What about if no prognostic biomarkers are available?

15)   Quoting from the work of Terada N et al., Ther Adv Med Oncol. 2017 Aug; 9(8): 565–573. "The presence or absence of a prognostic marker can be useful for the selection of patients for treatment but does not directly predict the response to treatment... PSA is organ- but not cancer-specific. Moreover, it is not able to differentiate between indolent and aggressive forms of prostate cancer. Many men may harbor aggressive prostate cancer despite having low initial levels of serum PSA (Thompson et al. 2004 N Engl J Med 2004; 350: 2239–2246) So can PSA always be an accurate biomarker for a cancer in clinical applications? In clinical applications we need prognosis or also prediction capabilities in cancer models? I think the authors mix these terms as they are presented (line 354).

16)   Author Contributions. I believe this work is one man show, TP only and the two other authors need to justify their involvement.

17)   Oncology is a continuous evolving field with novel insights and parameters to be introduced continuously. How can a researcher ever be sure that all the parameters are there in the model so it is identifiable?

Generally, I believe that the problem is not so clearly stated. Although the issue of identifiability is interesting to be introduced it should have been better addressed as a potential consideration in tumor growth models as a mean to validate or characterize how much applicable can a model be. Overall, I think this is a theoretical research work which can be acceptable after major revisions.

Author Response

Please see attached response.

Reviewer 2 Report

Manuscript: Practical understanding of model identifiability in clinical applications

By: Phan et al

I found this study interesting and relevant in the field.

The article is well written.

I found the conclusion to be in line with the evidence and arguments presented.

The figures and tables are okay.

I have a few suggestions to improve their manuscript.

The abstract is a bit confusing.

The authors should cite more studies in the Introduction section about mathematical modeling such as PMID: 21576488, PMID: 32987354…etc

The authors should tone down lines between 381-382.

Author Response

Please see attached response.

Reviewer 3 Report

This manuscript presents an illuminating scientific contribution addressing the important issue of mathematical modeling of biological processes in the nowadays era of computational oncology.

Comment:

-   Figure 3: To me, part (c) shows a positive correlation. Please, recheck the legend.

Author Response

Please see attached response.

Round 2

Reviewer 1 Report

The authors presented an updated version of their work clarifying most of the comments. Some additional observations: 

1) Clinical observations form the bridge between the mathematical models and application. No they do not. I insist, please, delete it. It is wrong. Models are the bridge between the two to help us understand some phenomena that take place. 

The idea of "models" in medicine is ~90 years old, maybe even before Turing's works (you may know better). See the work of Teorell T. Kinetics of distribution of substances administered to the body. I. The extravascular modes of administration. Arch Int Pharmacodyn Ther. 1937;57:205–225. 

Mathematical equations often helped clinicians to understand or describe clinical observations. Not the other way around. I provide relevant references for "bottom-up" approaches or how mathematical models (even in cancer) can be used to predict clinical observations or assist in explaining clinical scenarios. The models are usefull tools to explain clinical observations towards application. For example, outliers. Also there are models that are used in clinical applications BUT are not based in clinical observations (even for outliers). See the literature of drug research and how in vitro/in vivo data are extrapolated to clinical scenarios, even informing drug labels! The authors write about models in general. So please, delete it. 

Br J Clin Pharmacol. 2015 Jan;79(1):48-55. doi: 10.1111/bcp.12234. 

Pharmaceuticals (Basel). 2022 Jun 27;15(7):796. doi: 10.3390/ph15070796.

AAPS J. 2013 Apr;15(2):377-87. doi: 10.1208/s12248-012-9446-2. Epub 2012 Dec 27.

Cell Death Dis 9, 810 (2018). https://doi.org/10.1038/s41419-018-0865-6

2) Regarding the graphical abstract, I believe it is necessary according to Journal's instructions for authors (see relevant page). I leave it to the editor to decide if it is needed or not in this case. I believe it is needed. 

3) Figure 1. Please change the colours of overlaping lines or use less, or other simulation scenarios with no overlaps. (I would use Figure 1 as graphical abstract to present my case). 

4) What about if no prognostic biomarkers are available? The question was about clinically established, widely accepted biomarkers such as PSA or biomarkers that are exploited before and after intervention as treatment guidance etc. I can understand the confusion from the authors. For example a biomarker like PSA is not always feasible (even PSA has its problems) but usually several biomarkes are incorporated as potential clinical utilization. For example in glioblastoma due to its molecular heterogeneity there are not any specific biomarkers established. But even then, models not only exploited theorhetically (as authors answer) but indeed in pre-clinical and clinical applications. Examples:

Front Oncol. 2020 Dec 8;10:604121. doi: 10.3389/fonc.2020.604121. eCollection 2020.; 

Am J Cancer Res. 2020 Aug 1;10(8):2242-2257. eCollection 2020.; 

Neuro Oncol. 2020 Aug 17;22(8):1138-1149. doi: 10.1093/neuonc/noaa091. 

And many, many, many other models about glioblastoma. But I get the point of view from the authors and in order to avoid any conflicts please edit the relevant parts stating "mathematical-based prognostic biomarkers" i.e. L145, L260

I believe the authors tried their best. I cannot decide wheather this is a major/minor revision because the text is improved, the content very interesting as a bioengineering issue, of interest for the readers and possible uuseful, but still these comments are needed to be adressed
